

# Comparison of injury pattern and clinical outcomes between young adults and elderly patients with alcohol-related injury in South Korea 2011–2016

Jae Hee Lee and  Duk Hee Lee

Department of Emergency Medicine, Ewha Womans University, Seoul, South Korea

## ABSTRACT

**Background**. Alcohol is an important factor that contributes to emergency department (ED) visits due to injury. However, the role of alcohol in elderly patients visiting ED due to injury has not been clearly defined. This study aims to examine age and alcohol as risk factors of injury severity and clinical outcomes.

**Methods**. This study included patients who visited EDs between January 2011 and December 2016. Data was obtained from the Emergency Department-Based Injury In-depth Surveillance of the Korea Centers for Disease Control and Prevention, South Korea. Injury patients aged ≥18 years were included, but those who visited the ED more than 48 hours after injury, with unknown clinical outcomes (admission, mortality, and excess mortality ratio-adjusted injury severity score [EMR-ISS]) were excluded.

**Results**. We analyzed 887,712 patients, of whom 131,708 (17.7%) non-elderly and 9,906 (7.0%) elderly had alcohol-related injury. Falls and slips are the most common injury mechanism (37.9%) in patients consuming alcohol (36.3% non-elderly/58.40% elderly). The injury occurred on roads (40.6%), houses (33.8%), and commercial facilities (11.9%) in elderly patients consuming alcohol. Suicide rate was 12.0% in elderly and 9.7% in non-elderly patients. According to the time of day of injury, evening (60.8%) was the most common in elderly and night (62.6%) in non-elderly patients. Admission rate (odds ratio [OR] 2.512 confidence interval [CI] 2.407–2.621), intensive care unit (ICU) care rate (OR 5.507 [CI] 5.178–5.858), mortality rate (OR 4.593 [CI] 4.086–5.162), and EMR-ISS >25 (OR 5.498 [CI] 5.262–5.745) were compared between patients with alcohol-related injury and non-elderly with non-alcohol-related injury patients. Alcohol consumption in elderly patients results in significant impairment and increases EMR-ISS, ICU care rate, and mortality rate. To reduce injury in elderly patients, alcohol screening, appropriate counseling, and intervention are needed.

Corresponding author
Duk Hee Lee, calla@ewha.ac.kr, ewhain78@gmail.com

## INTRODUCTION

Injury is a public health problem accounting for 16% of the global disease burden. Alcohol is an important contributing factor to injury-related emergency department (ED) visits and is the most commonly used and abused substance in the United States. It accounts for one in 10 deaths among adults aged 20–64 years (*Fernandez et al., 2019*). Several studies have

suggested that alcohol consumption is independently associated with injury complications and severity (*Draus Jr et al., 2008*; *Plurad et al., 2011*; *Rehm et al., 2003*).

South Korea has a rapidly aging population. In 2015, 13.2% of South Korea's population was determined to be ≥65 years old. *KOSIS (2016)* Park reported that age affects fatality in injury patients (*Park et al., 2016*). The importance of the association between alcohol use and injury in elderly patients has been recognized (*Kowalenko et al., 2013*; *Selway, Soderstrom & Kufera, 2008*).

The role of alcohol in injury patients visiting EDs has not been clearly defined. This study aimed to examine the association of age and alcohol with injury severity and clinical outcomes.

## MATERIALS AND METHODS

### Study design and setting

This is a retrospective observational study on ED-based injury in-depth surveillance data obtained from the Korea Centers for Disease Control and Prevention, South Korea, which has been prospectively gathering injury-related information nationwide. During the study period, 20 EDs from 2011 to 2014 and 23 EDs from 2015 to 2016 participated in the survey. The hospitals are tertiary academic teaching hospitals across the nation. The data were anonymized before analysis. The Korea Centers for Disease Control and Prevention sends researchers to each hospital who can manage and checkup the data and offer continuous education programs to maintain data quality.

### Study population

This study population included patients who visited EDs between January 2011 and December 2016. Injury patients aged ≥18 years old were included. In South Korea, drinking alcohol is prohibited for those aged <18 years. Therefore, we excluded patients aged <17 years and are not known to be drinking. We defined elderly patients as those aged ≥65 years.

### Data collection and variables

The following variables were collected from electronic medical records based on the definition of variables provided by the Korea Centers for Disease Control and Prevention. We analyzed the following data: patient demographics (age and sex), drinking status (alcohol vs non-alcohol), mode of arrival (walk-in, 119, private ambulance, police, or air transportation), type of insurance (national health insurance, self-pay, vehicle, medicaid beneficiary, private insurance, or work accident), injury-related data (mechanism, place, kind of activity, intentionality, time, and day). Times of ED visit were classified into three categories: day (07:00 to 14:59), evening (15:00 to 22:59), and night (23:00 to 6:59).

Study outcomes were clinical results, rate of admission, intensive care unit (ICU) care, mortality, excess mortality ratio-adjusted injury severity score (EMR-ISS).

### Statistical analysis

Cross-tabulation was performed for non-continuous variables, and Student's *t*-test was used for continuous variables. A *P*-value of <0.05 was considered statistically significant.

To analyze variables associated among alcohol status, age, and outcomes (admission, ICU care, mortality, and EMR-ISS), univariate and multivariate logistic regression analyses were used.

## Ethics statement

This study was approved by the institutional review board (IRB) of Ewha Womans' University Mok-dong hospital (IRB No. 2019-05-030), and informed consent was waived by the IRB because patient information was anonymized before the analysis.

## RESULTS

During the study period, there were 1,537,617 injured patients in the registry, of whom 54,492 patients who visited the ED more than 48 h after injury were excluded; the drinking status of 87,856 patients were not known, and 498,969 were aged <18 years. We also excluded patients who were unknown whether they were admitted to hospital (1,921 patients) and whose EMR-ISS (6,667 patients) were unknown. Finally, we analyzed the data of 887,712 patients. Figure 1 shows the study flow diagram of enrolled patients.

### Demographic data of injury patients at ED in the elderly and non-elderly groups during 2011–2016 (Table 1)

Table 1 shows demographics of injury patients who visited ED during 2011–2016. There were 745,983 (84.0%) non-elderly patients (mean age 40.00 ± 12.95) and 141,729 (16.0%) elderly patients (mean age 74.67 ± 7.04). The number of male patients was 458,594 (61.4%) in non-elderly and 66,481 (46.9%) in elderly patients. The mode of ED visit in non-elderly patients was walk-in in 516,067 (69.2%), 119 in 184,353 (24.7%), and private ambulance in 41,840 (5.6%) patients, where in elderly patients, it was walk-in in 65,389 (46.1%), 119 in 53,782 (37.9%), and private ambulance in 21,845 (15.4%). The proportion of patients who used national health insurance was 75.4%, vehicle insurance was 15.2%, and medicaid beneficiary was 3.3% in injury patients; 141,614 (16.0%) patients who visited the ED due to injury consumed alcohol, and 131,708 (17.7%) of non-elderly and 9,906 (7.0%) of elderly patients had alcohol-related injury.

### Injury characteristics of patients at the ED in the elderly and non-elderly groups during 2011–2016 (Table 2)

Table 2 compares the mechanism, place, kind of activity, intentionality, and time of ED presentation between the non-elderly and elderly groups. In injury mechanism, the rate of falls and slips was significantly higher in the elderly (51.4%) than in the non-elderly (22.6%) group. The rate of collision (21.5%) and penetration (14.5%) injuries in non-elderly is higher than that in elderly patients (8.5% and 6.1%, respectively). In elderly patients, the most common place of injury was in the house (44.2%); furthermore, the rate of injury in commercial facilities (13.7% vs 5.1%) and factory, industrial, and construction facilities (8.1% vs 2.0%) was higher in non-elderly than in elderly patients. Elderly patients sustained injuries with daily living activities (49.5%) and leisure activities (17.5), whereas non-elderly patients sustained injuries with daily living activities (32.0%), leisure activities (20.2%), and work (16.3%).

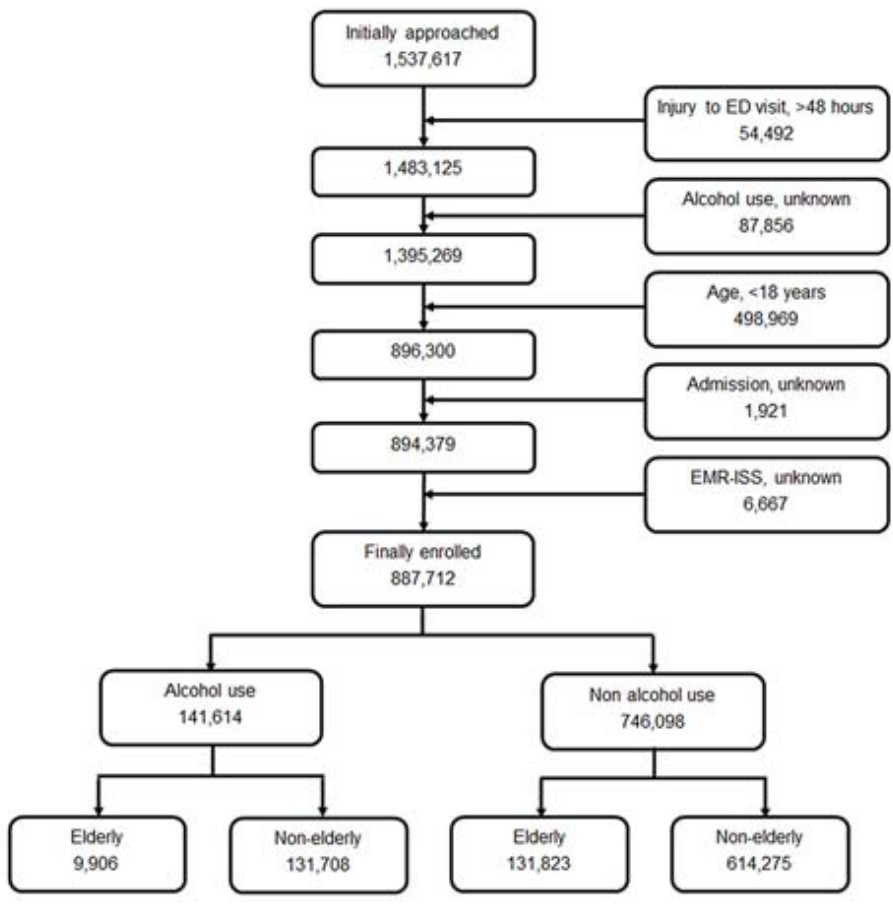

**Figure 1  Study flow diagram of enrolled patients.**

### General characteristics of injury patients in the elderly and non-elderly groups who consumed alcohol (Table 3)

We analyzed injury patients who consumed alcohol. There were 74.6% males (73.8% non-elderly/85.70% elderly). These patients had a higher rate of visiting EDs by 119 ambulance (43.9% non-elderly/61.0% elderly). Around 5.0% availed medicaid beneficiary (4.8% non-elderly/8.0% elderly).

### Injury characteristics in the elderly and non-elderly groups who consumed alcohol (Table 4, Figs. 2 and 3)

Table 4 compares the mechanism, place, kind of activity, intentionality, and time of ED presentation between non-elderly and elderly group who consumed alcohol. Fall and slips are the most common injury mechanism (37.9%) in these patients (36.3% non-elderly/58.40% elderly).

   Place of injury was the roads (40.6%), around the house (33.8%), and commercial facilities (11.9%) in elderly patients and roads (36.8%), commercial facility (31.3%), and around the house (21.6%) in non-elderly patients. Activities that led to injury was leisure activities (35.2%) and daily living activities (28.2%) in elderly patients and leisure activities
**Table 1  Demographic data of injury patients at ED in the elderly and non-elderly groups during 2011–2016.**

|  | Non-elderly (<65 yrs) | Elderly (≥65 yrs) | Total | p-value |
|---|---|---|---|---|
| No. of patients | 745,983 (84.0) | 141,729 (16.0) | 887,712 (100) | |
| Age (yrs, mean ± SD) | 40.0 ± 12.95 | 74.67 ± 7.04 | 45.54 ± 17.61 | <0.001 |
| Sex | | | | <0.001 |
|     Male | 458,594 (61.5) | 66,481 (46.9) | 525,075 (59.1) | |
|     Female | 287,389 (38.5) | 75,248 (53.1) | 362,637 (40.9) | |
| Mode of arrival | | | | |
|     Walk-in (include car, foot, etc.) | 516,067 (69.2) | 65,389 (46.1) | 581,456 (65.5) | <0.001 |
|     119 | 184,353 (24.7) | 53,782 (37.9) | 238,135 (26.8) | <0.001 |
|     Private ambulance | 41,840 (5.6) | 21,845 (15.4) | 63,694 (7.2) | <0.001 |
|     Police | 1,720 (0.2) | 108 (0.1) | 1,828 (0.2) | <0.001 |
|     Air | 1,204 (0.2) | 332 (0.2) | 1,536 (0.2) | <0.001 |
|     Others | 638 (0.1) | 230 (0.2) | 868 (0.1) | <0.001 |
|     Unknown | 161 (0.0) | 34 (0.0) | 195 (0.0) | 0.575 |
| Insurance | | | | |
|     National health insurance | 559,992 (75.1) | 109,145 (77.0) | 669,137 (75.4) | <0.001 |
|     Self-pay (uninsured) | 44,334 (5.9) | 4,591 (3.2) | 48,925 (5.5) | <0.001 |
|     Vehicle | 115,967 (15.5) | 19,127 (13.5) | 135,094 (15.2) | <0.001 |
|     Medicaid beneficiary | 20,481 (2.7) | 8,474 (6.0) | 28,955 (3.3) | <0.001 |
|     Private insurance | 132 (0.0) | 25 (0.0) | 157 (0.0) | 0.989 |
|     Work accident | 2,595 (0.3) | 185 (0.1) | 2,780 (0.3) | <0.001 |
|     Others | 2,268 (0.3) | 170 (0.1) | 2,438 (0.3) | <0.001 |
|     Unknown | 214 (0.0) | 12 (0.0) | 226 (0.0) | <0.001 |
| Alcohol ingestion before injury | | | | <0.001 |
|     Yes | 131,708 (17.7) | 9,906 (7.0) | 141,614 (16.0) | |
|     No | 614,275 (82.3) | 131,823 (93.0) | 746,098 (84.0) | |

(31.8%) and daily living activities (19.6%) in non-elderly patients. On the intentionality of injury, self-harm and suicide rates were 12.0% in elderly and 9.7% in non-elderly patients.

Injury was more common in weekdays (52.8%) in elderly patients and in weekends (52.9%) in non-elderly patients. The monthly incidence of injury showed no significant difference between elderly and non-elderly patients (Fig. 2). With respect to the time of injury, evening (60.8%) was the most common in elderly and night (62.6%) in non-elderly patients (Fig. 3).

## Comparison of clinical outcomes and injury severity between non-elderly and elderly patients by alcohol use (Table 5)

We divided the study group by age and alcohol use. In the alcohol use group, the elderly had a higher ICU care rate (odds ratio [OR] 3.065 confidence interval [CI] 2.872–3.727), mortality rate (OR 5.136 [CI] 4.501–5.861), and EMR-ISS (≥25) (OR 2.535 [CI] 2.249–2.462). In the non-alcohol use group, the elderly had a higher ICU care rate (OR 2.709

**Table 2 Injury characteristics of patients at the ED in the elderly and non-elderly groups during 2011–2016.**

| | Non-elderly (<65 yrs) | Elderly (≥65 yrs) | Total | p-value |
|---|---|---|---|---|
| **Mechanism** | | | | |
| Fall, slip | 168,255 (22.6) | 72,914 (51.4) | 241,169 (27.2) | <0.001 |
| Collision | 160,157 (21.5) | 12,041 (8.5) | 172,199 (19.4) | <0.001 |
| Traffic accident | 160,190 (21.5) | 28,726 (20.3) | 188,916 (21.3) | <0.001 |
| Penetration | 108,135 (14.5) | 8,581 (6.1) | 116,716 (13.1) | <0.001 |
| Substance exposure | 26,521 (3.6) | 6,347 (4.5) | 32,868 (3.7) | <0.001 |
| Overuse | 29,624 (4.0) | 2,916 (2.1) | 32,540 (3.7) | <0.001 |
| Drowning, hanging, asphyxia | 2,562 (0.3) | 814 (0.6) | 3,376 (0.4) | <0.001 |
| Thermal injury | 17,130 (2.3) | 1,218 (0.9) | 18,348 (2.1) | <0.001 |
| Machine | 9,570 (1.3) | 883 (0.6) | 10,453 (1.2) | <0.001 |
| Natural disaster | 51 (0.0) | 15 (0.0) | 66 (0.0) | 0.134 |
| Others | 66,010 (8.0) | 6,328 (4.5) | 66,338 (7.5) | <0.001 |
| Unknown | 3,778 (0.5) | 945 (0.7) | 4,723 (0.5) | <0.001 |
| **Place** | | | | |
| Road | 239,594 (32.1) | 44,567 (31.4) | 284,161 (32.0) | <0.001 |
| Commercial facilities | 102,419 (13.7) | 7,220 (5.1) | 109,639 (12.4) | <0.001 |
| House | 209,573 (28.1) | 62,614 (44.2) | 272,187 (30.7) | <0.001 |
| Outdoor, river, sea | 34,180 (4.6) | 6,931 (4.9) | 41,111 (4.6) | <0.001 |
| Amusement, cultural public facilities | 13,473 (1.8) | 2,377 (1.7) | 15,850 (1.8) | 0.001 |
| Transportation area except road | 9,504 (1.3) | 2,724 (1.9) | 12,228 (1.4) | <0.001 |
| Residential facilities | 6,696 (0.9) | 2,716 (1.9) | 9,412 (1.1) | <0.001 |
| Factory, industrial facilities | 60,766 (8.1) | 2,803 (2.0) | 63,569 (7.2) | <0.001 |
| Farm | 8,388 (1.1) | 3,442 (2.4) | 11,830 (1.3) | <0.001 |
| School, education facilities | 6,633 (0.9) | 160 (0.1) | 6,793 (0.8) | <0.001 |
| Sport facilities | 32,876 (4.4) | 981 (0.7) | 33,857 (3.8) | <0.001 |
| Medical facilities | 12,059 (1.6) | 3,879 (2.7) | 15,938 (1.8) | <0.001 |
| Others | 641 (0.1) | 134 (0.1) | 775 (0.1) | <0.001 |
| Unknown | 9,181 (1.2) | 1,811 (0.8) | 10,362 (1.2) | <0.001 |
| **Activity** | | | | |
| Leisure activities | 150,961 (20.2) | 24,865 (17.5) | 175,726 (19.8) | <0.001 |
| Daily living activities | 238,731 (32.0) | 70,129 (49.5) | 308,860 (34.8) | <0.001 |
| Unpaid labor | 103,240 (13.8) | 23,729 (16.7) | 126,969 (14.3) | <0.001 |
| Work | 121,689 (16.3) | 10,258 (7.2) | 131,947 (14.9) | <0.001 |
| Exercise | 35,111 (4.7) | 1,785 (1.3) | 36,896 (4.2) | <0.001 |
| Travel | 3,419 (0.5) | 453 (0.3) | 3,872 (0.4) | <0.001 |
| Hospital treatment | 2,036 (0.3) | 1,863 (1.3) | 3,899 (0.4) | <0.001 |
| Education | 2,739 (0.4) | 27 (0.0) | 2,766 (0.3) | <0.001 |
| Others | 83,212 (11.2) | 7,675 (5.4) | 90,884 (10.2) | <0.001 |
| Unknown | 4,845 (0.6) | 1,045 (0.7) | 5,890 (0.7) | <0.001 |

**Table 2** (*continued*)

| | Non-elderly (<65 yrs) | Elderly (≥65 yrs) | Total | *p*-value |
|---|---|---|---|---|
| Intentionality | | | | |
|     Unintentional | 664,108 (89.0) | 134,390 (94.8) | 798,498 (90.0) | <0.001 |
|     Assault | 53,541 (7.2) | 2,078 (1.5) | 55,619 (6.3) | <0.001 |
|     Self-harm, suicide | 25,167 (3.4) | 4,500 (3.2) | 29,667 (3.3) | <0.001 |
|     Others | 906 (0.1) | 204 (0.1) | 1,110 (0.1) | 0.028 |
|     Unknown | 2,261 (0.3) | 557 (0.4) | 2,818 (0.3) | <0.001 |
| Day of presentation | | | | <0.001 |
|     Weekday (Mon-Thu) | 355,599 (47.7) | 75,902 (53.6) | 431,501 (48.6) | |
|     Weekend (Fri-Sun) | 390,384 (52.3) | 65,827 (46.4) | 456,211 (51.4) | |
| Time of presentation | | | | |
|     Day (7∼14 h) | 204,742 (27.4) | 59,076 (41.7) | 263,818 (29.7) | <0.001 |
|     Evening (15∼22 h) | 335,621 (45.0) | 66,125 (46.7) | 401,746 (45.3) | <0.001 |
|     Night (23∼6 h) | 205,606(27.6) | 16,526 (11.7) | 222,132 (25.0) | <0.001 |

**Table 3** General characteristics of injury patients in the elderly and non-elderly groups who consumed alcohol.

| | Non-elderly (<65 yrs) | Elderly (≥65 yrs) | Total | *p*-value |
|---|---|---|---|---|
| No. of patients | 131,708 (93.0) | 9,906 (7.0) | 141,614 (100) | |
| Age | 38.98 ± 12.62 | 71.50 ± 5.36 | 41.25 ± 14.80 | <0.001 |
| Sex | | | | <0.001 |
|     Male | 97,176 (73.8) | 8,493 (85.7) | 105,669 (74.6) | |
|     Female | 34,532 (26.2) | 1,413 (14.3) | 35,945 (25.4) | |
| Mode of arrival | | | | |
|     Walk-in (include car, foot, etc.) | 65,227 (49.5) | 2,524 (25.5) | 67,751 (47.8) | <0.001 |
|     119 | 57,773 (43.9) | 6,047 (61.0) | 63,820 (45.1) | <0.001 |
|     Private ambulance | 7,570 (5.7) | 1,280 (12.9) | 8,850 (6.2) | <0.001 |
|     Police | 925 (0.7) | 28 (0.3) | 953 (0.7) | <0.001 |
|     Air | 93 (0.1) | 19 (0.2) | 112 (0.1) | <0.001 |
|     Others | 88 (0.1) | 5 (0.1) | 93 (0.1) | 0.540 |
|     Unknown | 32 (0.0) | 3 (0.0) | 35 (0.0) | 0.715 |
| Insurance | | | | |
|     National health insurance | 103,975 (78.9) | 8,064 (81.4) | 112,039 (79.1) | <0.001 |
|     Self-pay (uninsured) | 12,702 (9.6) | 596 (6.0) | 13,298 (9.4) | <0.001 |
|     Vehicle | 8,193 (6.2) | 434 (4.4) | 8,627 (6.1) | <0.001 |
|     Medicaid beneficiary | 6,299 (4.8) | 789 (8.0) | 7,088 (5.0) | <0.001 |
|     Private insurance | 24 (0.0) | 3 (0.0) | 27 (0.0) | 0.402 |
|     Work accident | 23 (0.0) | 1 (0.0) | 24 (0.0) | 0.587 |
|     Others | 416 (0.3) | 16 (0.2) | 432 (0.3) | 0.007 |
|     Unknown | 76 (0.1) | 3 (0.0) | 79 (0.1) | 0.265 |

**Table 4  Injury characteristics in the elderly and non-elderly groups who consumed alcohol.**

|  | Non-elderly (<65 yrs) | Elderly (≥65 yrs) | Total | *p*-value |
|---|---|---|---|---|
| **Mechanism** |  |  |  |  |
| Fall, slip | 47,861 (36.3) | 5,789 (58.4) | 53,650 (37.9) | <0.001 |
| Collision | 41,992 (31.9) | 1,048 (10.6) | 43,040 (30.4) | <0.001 |
| Traffic accident | 13,658 (10.4) | 1,073 (10.8) | 14,731 (10.4) | 0.146 |
| Penetration | 12,576 (9.5) | 238 (2.4) | 12,814 (9.0) | <0.001 |
| Substance exposure | 9,077 (6.9) | 1,292 (13.0) | 10,369 (7.3) | <0.001 |
| Overuse | 1,236 (0.9) | 33 (0.3) | 1,269 (0.9) | <0.001 |
| Drowning, hanging, asphyxia | 836 (0.6) | 62 (0.6) | 898 (0.6) | 0.915 |
| Thermal injury | 751 (0.6) | 18 (0.2) | 769 (0.5) | <0.001 |
| Machine | 60 (0.0) | 8 (0.1) | 68 (0.0) | 0.123 |
| Natural disaster | 5 (0.0) | 0 (0.0) | 5 (0.0) | 0.540 |
| Others | 1,499 (1.1) | 71 (0.7) | 1,570 (1.1) | <0.001 |
| Unknown | 2,157 (1.6) | 274 (2.8) | 2,431 (1.7) | <0.001 |
| **Place** |  |  |  |  |
| Road | 48,487 (36.8) | 4,026 (40.6) | 52,513 (37.1) | <0.001 |
| Commercial facilities | 41,254 (31.3) | 1,176 (11.9) | 42,430 (30.0) | <0.001 |
| House | 28,506 (21.6) | 3,349 (33.8) | 31,855 (22.5) | <0.001 |
| Outdoor, river, sea | 3,434 (2.6) | 342 (3.5) | 3,776 (2.7) | <0.001 |
| Amusement, cultural public facilities | 2,533 (1.9) | 200 (2.0) | 2,733 (1.9) | 0.504 |
| Transportation area except road | 1,747 (1.3) | 378 (3.8) | 2,125 (1.5) | <0.001 |
| Residential facilities | 1,127 (0.9) | 119 (1.2) | 1,246 (0.9) | <0.001 |
| Factory, industrial facilities | 591 (0.4) | 28 (0.3) | 619 (0.4) | 0.016 |
| Farm | 298 (0.2) | 50 (0.5) | 348 (0.2) | <0.001 |
| School, education facilities | 322 (0.2) | 3 (0.0) | 325 (0.2) | <0.001 |
| Sport facilities | 302 (0.2) | 12 (0.1) | 314 (0.2) | 0.027 |
| Medical facilities | 157 (0.1) | 36 (0.4) | 193 (0.1) | <0.001 |
| Others | 99 (0.1) | 9 (0.1) | 108 (0.1) | 0.585 |
| Unknown | 2,851 (2.2) | 178 (1.8) | 3,029 (2.1) | 0.015 |
| **Activity** |  |  |  |  |
| Leisure activities | 41,853 (31.8) | 3,486 (35.2) | 45,339 (32.0) | <0.001 |
| Daily living activities | 25,840 (19.6) | 2,789 (28.2) | 28,629 (20.2) | <0.001 |
| Unpaid labor | 13,987 (10.6) | 1,235 (12.5) | 15,222 (10.7) | 0.129 |
| Work | 1,345 (1.0) | 117 (1.2) | 1,462 (1.0) | <0.001 |
| Exercise | 286 (0.2) | 10 (0.1) | 296 (0.2) | 0.015 |
| Travel | 187 (0.1) | 8 (0.1) | 195 (0.1) | 0.113 |
| Hospital treatment | 53 (0.0) | 15 (0.2) | 68 (0.0) | <0.001 |
| Education | 48 (0.0) | 3 (0.0) | 51 (0.0) | 0.755 |
| Others | 46,538 (35.3) | 2,053 (20.7) | 48,591 (34.3) | <0.001 |

**Table 4** (*continued*)

|  | Non-elderly (<65 yrs) | Elderly (≥65 yrs) | Total | *p*-value |
|---|---|---|---|---|
| Unknown | 1,571 (1.2) | 190 (1.9) | 1761 (1.2) | <0.001 |
| Intentionality |  |  |  |  |
| Unintentional | 86,063 (65.3) | 7,913 (79.9) | 93,976 (66.4) | <0.001 |
| Assault | 31,201 (23.7) | 618 (6.2) | 31,819 (22.5) | <0.001 |
| Self-harm, suicide | 12,825 (9.7) | 1,192 (12.0) | 14,017 (9.9) | <0.001 |
| Others | 328 (0.2) | 37 (0.4) | 365 (0.3) | 0.018 |
| Unknown | 1,291 (1.0) | 146 (1.5) | 1,437 (1.0) | <0.001 |
| Day of presentation |  |  |  | <0.001 |
| Weekday (Mon-Thu) | 63,372 (48.1) | 5,233 (52.8) | 68,605 (48.4) |  |
| Weekend (Fri-Sun) | 68,336 (51.9) | 4,673 (47.2) | 73,009 (51.6) |  |
| Time of presentation |  |  |  |  |
| Day (7~14 h) | 17,291 (13.1) | 1,555 (15.7) | 18,846 (13.3) | <0.001 |
| Evening (15~22 h) | 31,941 (24.3) | 6,019 (60.8) | 37,960 (26.8) | <0.001 |
| Night (23~6 h) | 82,468 (62.6) | 2,332 (23.5) | 141,606 (59.9) | <0.001 |

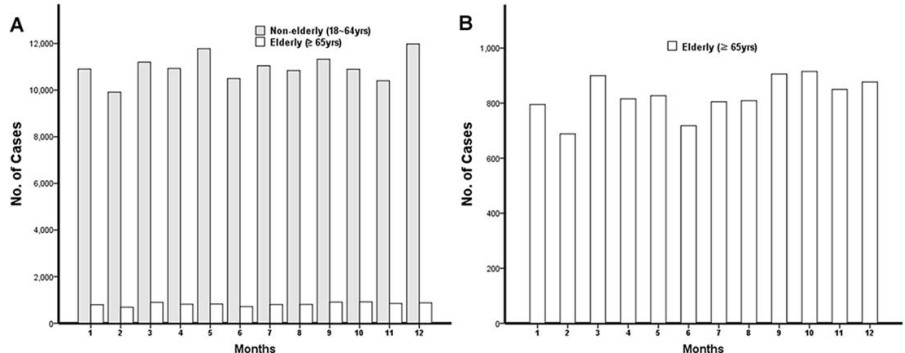

**Figure 2 Number of patients who visit ED by month.** (A) The monthly injury incidence of non-elderly and elderly patients. (B) The monthly injury incidence of elderly patients.

[CI] 2.637–2.783), mortality rate (OR 4.456 [CI] 4.266–4.653), and EMR-ISS (≥25) (OR 2.541 [CI] 2.497–2.538).

## Logistic regression for interaction between clinical outcomes and age and alcohol use in injured patients (Table 6)

Statistically significant interaction effect was seen between age and alcohol use. To evaluate the effect of age and alcohol, we divided the study population into non-elderly, non-alcohol use group (G1), non-elderly, alcohol use group (G2), elderly, non-alcohol use group (G3), and elderly, alcohol use group (G4). Logistic regression revealed statistically significant differences among the four groups in admission rate, ICU care rate, mortality rate, and EMR-ISS. Table 6 shows ORs and 95% CI with G1 as the reference. Compared with G1, the OR and CI in G2, G3, and G4 were OR 1.045 [CI] 1.028–1.062, OR 3.499 [CI]

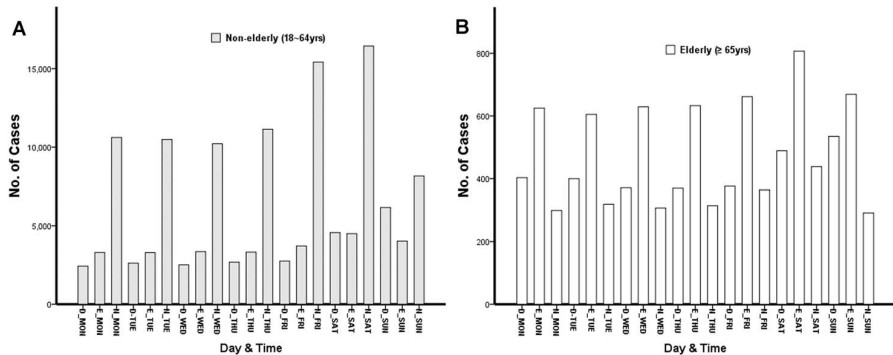

**Figure 3 Number of patients who visit ED by day and time.** (A) The time of injury incidence in non-elderly patients. (B) The time of injury incidence in elderly patients.

**Table 5 Comparison of clinical outcomes and injury severity between non-elderly and elderly patients by alcohol use.**

|  | Non-elderly (<65 yrs) | Elderly (≥65 yrs) | *p*-value | OR (95% CI) |
|---|---|---|---|---|
| **Alcohol use** |  |  |  |  |
| Admission | 22,091 (16.8) | 3,233 (32.6) | <0.001 | 2.404 (2.300–2.513) |
| ICU | 5,853 (4.4) | 1,236 (12.5) | <0.001 | 3.065 (2.872–3.272) |
| Mortality | 826 (0.6) | 311 (3.1) | <0.001 | 5.136 (4.501–5.861) |
| EMR-ISS (≥25) | 20,568 (15.6) | 3,005 (30.3) | <0.001 | 2.353 (2.249–2.462) |
| **No alcohol use** |  |  |  |  |
| Admission | 99,317 (16.2) | 53,117 (40.3) | <0.001 | 3.499 (3.454–3.545) |
| ICU | 15,500 (2.5) | 8,638 (6.6) | <0.001 | 2.709 (2.637–2.783) |
| Mortality | 4,305 (0.7) | 4,019 (3.0) | <0.001 | 4.456 (4.266–4.653) |
| EMR-ISS (≥25) | 45,080 (7.3) | 22,081 (16.8) | <0.001 | 2.541 (2.497–2.585) |

3.454–3.545, and OR 2.512 [CI] 2.407–2.621, respectively, for admission rate; OR 1.797 [CI] 1.742–1.853, OR 2.709 [CI] 2.637–2.783, and OR 5.507 [CI] 5.178–5.858, respectively, for ICU care rate; OR 0.894 [CI] 0.830–0.964, OR 4.456 [CI] 4.266–4.653, and OR 4.593 [CI] 4.086–5.162, respectively, for mortality rate; OR 2.337 [CI] 2.296–2.378, OR 2.541 [CI] 2.497–2.585, and OR 5.498 [CI] 5.262–5.745, respectively, for EMR-ISS >25(*Kim et al., 2009*). G4 had higher OR for EMR-ISS, ICU care rate, and mortality rate.

## DISCUSSION

This study shows that alcohol-related injury had serious clinical outcomes in elderly patients. The relationship between alcohol consumption and injury has already been revealed by various studies and analyses. Acute alcohol consumption, even in small amounts, increases the risk of injury (*Cherpitel, 2007*). Globally, 10%–18% of injuries that result in ED visits have been reported to be related to alcohol consumption (*WHO, 2007*). In the elderly population, injury has important public health implications. Injured elderly patients have a higher mortality rate than younger adults (*Champion et al., 1989*), and it has

**Table 6 Logistic regression for interaction between clinical outcomes and age and alcohol use in injured patients.**

|  | OR | 95% CI | *p*-value |
|---|---|---|---|
| **Admission** | | | |
| Non-elderly, no alcohol use | 1.000 | Reference | – |
| Non-elderly, alcohol use | 1.045 | 1.028–1.062 | <0.001 |
| Elderly, no alcohol use | 3.499 | 3.454–3.545 | <0.001 |
| Elderly, alcohol use | 2.512 | 2.407–2.621 | <0.001 |
| **ICU** | | | |
| Non-elderly, no alcohol use | 1.000 | Reference | – |
| Non-elderly, alcohol use | 1.797 | 1.742–1.853 | <0.001 |
| Elderly, no alcohol use | 2.709 | 2.637–2.783 | <0.001 |
| Elderly, alcohol use | 5.507 | 5.178–5.858 | <0.001 |
| **Mortality** | | | |
| Non-elderly, no alcohol use | 1.000 | Reference | – |
| Non-elderly, alcohol use | 0.894 | 0.830–0.964 | 0.003 |
| Elderly, no alcohol use | 4.456 | 4.266–4.653 | <0.001 |
| Elderly, alcohol use | 4.593 | 4.086–5.162 | <0.001 |
| **EMR-ISS ($\geq$25)** | | | |
| Non-elderly, no alcohol use | 1.000 | Reference | – |
| Non-elderly, alcohol use | 2.337 | 2.296–2.378 | <0.001 |
| Elderly, no alcohol use | 2.541 | 2.497–2.585 | <0.001 |
| Elderly, alcohol use | 5.498 | 5.262–5.745 | <0.001 |

been reported that relatively mild mechanisms may cause injury in the elderly (*Giannoudis et al., 2009*). This study aimed to investigate the relationship between alcohol consumption and injury in elderly patients and to analyze the effect of alcohol on injury severity.

The rate of alcohol consumption among elderly patients with injury was 7.0% (Table 1). *Ekeh et al. (2014)* reported that 11.1% of injury patients aged $\geq$65 years who visited the level 1 trauma center showed positive blood alcohol content (BAC). In another study, the BAC of 12.6% of injury patients aged $\geq$65 years who visited the ED was >100 mg/dl. (*Rivara et al., 1993*) The difference between previous studies and the present study is probably due to the different alcohol-drinking behaviors of each region and cultures.

The number of males in the alcohol injury elderly group was higher than that in the younger adult group (Table 3). This is thought to be due to differences in alcohol-drinking behavior based on age and sex. According to Kim's analysis of data from the Korea National Health and Nutrition Examination Survey, prevalence of intermediate risk drinking (alcohol use disorder identification test score of $\geq$8) was 23.1%, 13.5%, and 4.2% for females aged 19–44 years, 45–64 years, and $\geq$65 years, respectively, and 59.7%, 60.1%, and 36.7%, respectively, for males (*Hong, Noh & Kim, 2017*). In elderly females, the drinking rate was significantly lower; this explains why the proportion of males in the elderly alcohol injury patients is significantly higher. It can also be seen that the rate of alcohol consumption is closely related to the occurrence of alcohol-related injury.

The rate of using 119 in the alcohol injury elderly group was higher than that in the total elderly injury group (Tables 1 and 3). A total of 119 of Korea, regardless of the severity, sends the patient to a hospital free of charge, if they request the emergency medical system. For this reason, 119 is more likely to be used when patients cannot move by themselves or in poor economic conditions. This tendency is prominent in the elderly. (*Kang, 2015*) Besides, the higher use of 119 in the drinking elderly group than in the total elderly injury group can be attributed to the increased severity of injury due to alcohol (Tables 4 and 5). At a time when the society is rapidly aging, this will lead to an increase in medical costs.

The percentage of falls and slips in the elderly alcohol group was higher than that in the total elderly injury group. Falls and slips are the most common and one of the major mechanisms in elderly injury. They can result in fractures, immobility, and reduced daily living activities, even leading to death (*WHO, 2017*). Risk factors for elderly falls include age, drug use, cognitive impairment, and sensory deficit (*Fuller, 2000*). Even though there are various opinions as to whether alcohol is a risk factor in elderly falls (*Adams & Jones, 1998*), studies have suggested that alcohol is a risk factor for the occurrence of and mortality due to elderly falls (*Schick et al., 2018*; *Sorock et al., 2006*). In this study, the incidence of falls and slips was higher in the elderly alcohol group, suggesting that alcohol contributes to the occurrence of falls.

In this research, intentional injury in patients who consumed alcohol was higher than in total injury patients. A total of 9.9% of patients who consumed alcohol had intentions of self-harm or suicide, whereas 12.0% of elderly patients who consumed alcohol attempted suicide and self-harm. *Darvishi et al. (2015)* reported that alcohol use significantly increases the risk of suicidal ideation, suicide attempt, and completed suicide. Compared with the total elderly injury group, intentional injury (assault, self-harm/suicide) accounted for a large proportion of the alcohol elderly injury group (Tables 2 and 4). Therefore, alcohol can be seen as one of the factors causing intentional injury. In the alcohol injury group, the ratio of unintentional injury and self-harm/suicide was higher in the elderly group than in the younger group. This is because of the higher rate of violence in the younger alcohol group. According to *Dinh et al. (2014)*, the proportion of alcohol-related assault patients who visited the trauma center was 3%, 52%, and 21% in those aged ≥65 years, 24–44 years, and 45–64 years old, respectively.

ED visits in elderly injury patients were the most frequent during the evening, followed by day, then night (Table 2). On the other hand, the order was evening, night, and day in elderly alcohol injury patients (Table 4). This is related to the time of day when alcohol drinking occurs. In the younger alcohol injury group, visits were the highest during the night, (Table 4) suggesting a difference in the time zone in which alcohol consumption is done in different age groups.

With regard to injury severity, age and drinking were found to have interaction effects. Injury severity was positively correlated with age and alcohol consumption. Particularly, this was observed in ICU care rate and EMR-ISS score of ≥25 in severely injured patients (Tables 5 and 6). Although there is no doubt that alcohol is a risk factor in causing damage, various studies have shown whether alcohol consumption is related to the severity of the damage. Some studies have reported that alcohol consumption and injury severity are

not significantly correlated (*Ahmed & Greenberg, 2019*; *Schneiders et al., 2017*). There was a difference according to the mechanism of damage. (*Valdez et al., 2016*) It has also been reported that the effect of alcohol is different depending on whether the BAC is 400 or more (*Afshar et al., 2016*). While positive BACs have been shown to reduce mortality rate in trauma patients (*Yaghoubian et al., 2009*), drinking has been reported to be related to injury severity (*Elshiere, Noorbhai & Madiba, 2017*; *Swearingen et al., 2010*). This study showed that age and drinking had an interaction effect and were associated with the occurrence of severe injury.

There are some limitations. This study used data obtained from the Emergency Department-Based Injury In-depth Surveillance conducted by the Korea Centers for Disease Control and Prevention at 23 EDs to prevent injury. However, one of the limitations of this study is that information on alcohol consumption behavior required to understand the relationship between injury and drinking is limited, and the exact time interval between alcohol consumption and injury is unknown. Additionally, the fact that alcohol consumption was based on the patient's self-report was a limitation to objectively determine whether the patient was drinking or not. In England, alcohol screening with referral or intervention is becoming part of routine practice. And the rate of alcohol identification and intervention in ED has had improvements (*Patton & Green, 2018*). However, there is no system for alcohol screening and intervention in Korea. 23 EDs supported by KCDC gather the information whether the injury patients drink or not. Korean EDs need to screen alcohol with appropriate tools such as AUDIT score for future study and public health.

Tests to reveal blood alcohol level were not performed. *Zautcke et al. (2002)* also reported that falls are the most common mechanism of injury in elderly patients who under the influence of alcohol. They performed blood alcohol level test in 5.2% of geriatric patients and obtained positive results for 49.7% (*Zautcke et al., 2002*). *Csipke et al. (2007)* reported a low correlation with BAC and alcohol questionnaire results. This suggests that not only alcohol screening but also an objective BAC measurement should be performed. Alcohol screening and BAC information should be included in the future study. And More prospective studies are needed to investigate true alcohol level. Also, there is a possibility of selection biases due to unknown data on alcohol use, admission rate, and EMR-ISS.

## CONCLUSIONS

Elderly alcohol injury patients increased the incidence of falls, the most common injury mechanism in elderly, and increased the utilization rate of 119. Moreover, age and alcohol were risk factors for ICU care and EMR-ISS score of $\geq 25$. Alcohol screening and intervention is required in geriatric trauma patients because alcohol use is significantly prevalent in trauma (*Ekeh et al., 2014*). This study shows that alcohol-related injury had serious clinical outcomes. Therefore, alcohol screening and intervention in ED are needed to prevent alcohol-related injury in the elderly.

## ACKNOWLEDGEMENTS

We are thanks to Hye Ah Lee. Ph.D. for helping statistical analysis.

### Funding

This work was supported by the National Research Foundation of Korea (NRF) grant funded by the Korea government (MSIT) (No. 2018R1C1B5046096), and a fund by Research of Korea Centers for Disease Control and Prevention (Comparison of injury pattern and clinical outcomes between young adults and elderly patients with alcohol-related injury in South Korea 2011–2016). The funders had no role in study design, data collection and analysis, decision to publish, or preparation of the manuscript.

### Grant Disclosures

The following grant information was disclosed by the authors:
National Research Foundation of Korea (NRF).
Korea government (MSIT): 2018R1C1B5046096.
Research of Korea Centers for Disease Control and Prevention.
South Korea: 2011–2016.

### Competing Interests

The authors declare there are no competing interests.

### Author Contributions

- Jae Hee Lee performed the experiments, analyzed the data, prepared figures and/or tables.
- Duk Hee Lee conceived and designed the experiments, performed the experiments, contributed reagents/materials/analysis tools, authored or reviewed drafts of the paper, approved the final draft.

### Human Ethics

The following information was supplied relating to ethical approvals (i.e., approving body and any reference numbers):

This study was approved by the institutional review board (IRB) of Ewha Womans' University Mok-dong hospital (IRB No.2019-05-030), and informed consent was waived by the IRB because patient information was anonymized before the analysis.

### Data Availability

The Korean Center for Disease Control (KCDC) is the authority for accessing the data analyzed, and there are ethical restrictions on sharing a dataset because the data contain potentially identifying information. The KCDC (http://www.cdc.go.kr) can be contacted for data access via the Injury research team email (kcdcinjury@korea.kr) or by calling 82-43-719-7407.

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
