# Peer review of "Comparison of injury pattern and clinical outcomes between young adults and elderly patients with alcohol-related injury in South Korea 2011–2016"

_PeerJ, doi:10.7717/peerj.7704_

## Round 0.1 · original submission · Minor Revisions

Thank you for your submission. This was well received by both reviewers. You should revise the abstract as suggested and providing additional detail as to the study design and methodology. Any further detail you can add with regard to levels of consumption would enhance your submission.

·

Basic reporting

This research study explores an area that is relevant to a cross-disciplinary readership and is of considerable relevance to clinical practice. It is coherent and succinct. The aims are clearly stated with structured data that is well organised. The results, discussion and limitations are clearly linked to the aims and methodology.

Experimental design

The methodology is robust, with a large data set interrogated to produce results that have sufficient power and significance to reduce the likelihood of false positive and negative results. Although limited by retrospective design and hence limited by ascertainment and observer bias, the sample size is representative, with the use of a control group against which to compare odds ratios of relative risk. Outcomes are valid and statistical analysis appropriate to experimental design.

Validity of the findings

The study is of high quality and fills a gap in knowledge surrounding the association between injury and alcohol use, with adequate reference to the literature,. It has considerable external validity, given that it draws on data obtained from clinical settings that could be conveniently replicated across other similar settings in other countries.

Additional comments

I found this paper of high quality, with robust aims, hypothesis, methodology, reporting of results and the discussion was balanced, qualified by limitations to the interpretation of results.

·

Basic reporting

The first paragraph ie the abstract part needs to be a synopsis of the paper ie defining and improving to make it more succinct and accurate:In the first paragraph, background rick should be risk. Not really understanding the methodology ie sentence:but those who visited the ED 48 h after injury, with unknown clinical outcomes. I'm assuming this was a retrospective study. Were the notes reviewed? Where did the information initially come from to get the patient group?

So I would suggest that the initial abstract is revised ie background and methods.


The above is clarified in the materials and methods in the main body of the paper. Was there any information about quantity of alcohol consumed? Any audit C data? This is still seen as the best tool for elderly alcohol consumption. Was there any information collected about poly pharmacy?

For a future study may also be worth breaking down the over 65s into age brackets.


Probably worth referencing the below:
Use of blood alcohol concentration in resuscitation room patients.

Csipke E1, Touquet R, Patel T, Franklin J, Brown A, Holloway P, Batrick N, Crawford MJ.

Experimental design

No comment. I thought the design was appropriate and any flaws were mentioned.

Validity of the findings

No comment

Additional comments

I would just suggesting revising the abstract part and perhaps reading the paper suggested to see if this could be added in somewhere.

---

## Round 0.2 · accepted · Accept

Thank you for your revised manuscript and accompanying comments. I am pleased to inform you that this paper has now been accepted for publication.